# *ESR1* NAPA Assay: Development and Analytical Validation of a Highly Sensitive and Specific Blood-Based Assay for the Detection of *ESR1* Mutations in Liquid Biopsies

**DOI:** 10.3390/cancers13030556

**Published:** 2021-02-01

**Authors:** Dimitra Stergiopoulou, Athina Markou, Eleni Tzanikou, Ioannis Ladas, G. Mike Makrigiorgos, Vassilis Georgoulias, Evi Lianidou

**Affiliations:** 1Analysis of Circulating Tumour Cells, Laboratory of Analytical Chemistry, Department of Chemistry, University of Athens, 15771 Athens, Greece; dimitrasterg@chem.uoa.gr (D.S.); etzanikou@chem.uoa.gr (E.T.); 2Department of Radiation Oncology, Dana-Farber Cancer Institute and Brigham and Women’s Hospital, Harvard Medical School, Boston, MA 02215-5450, USA; ladasivf@gmail.com (I.L.); mike_makrigiorgos@dfci.harvard.edu (G.M.M.); 3First Department of Medical Oncology, Metropolitan General Hospital, 155 62 Athens, Greece; georgulv@otenet.gr; 4Department of Medical Oncology, School of Medicine, University of Crete, 71500 Crete, Greece

**Keywords:** ER+ breast cancer, liquid biopsy, CTCs, cell free DNA, circulating tumour DNA, *ESR1*, mutation analysis, NAME-Pro, ARMS-PCR

## Abstract

**Simple Summary:**

A considerable number of estrogen-receptor–positive (ER+) breast cancer patients develop resistance to endocrine treatment. One of the most important resistance mechanisms is the presence of *ESR1* mutations. In the present study, we developed and analytically validated a novel, highly sensitive and specific nuclease-assisted minor-allele enrichment with probe-overlap (NaME-PrO)-assisted Amplification refractory mutation system (ARMS) (NAPA) assay for the detection of four *ESR1* mutations (Y537S, Y537C, Y537N and D538G). The assay was further applied in 13 ER+ breast cancer (BrCa) primary tumour tissues (FFPEs), 13 non-cancerous breast tissues (mammoplasties), and 32 pairs of liquid biopsy samples [circulating tumour cells (CTCs) and paired plasma circulating tumour DNA (ctDNA)] obtained at different time points from 8 ER+ metastatic breast cancer patients. In the plasma ctDNA, the *ESR1* mutations were not identified at the baseline, whereas the D538G mutation was detected during the follow-up period at five consecutive time points in one patient. In the CTCs, only the Y537C mutation was detected in one patient sample at the baseline. A direct comparison of the *ESR1* NAPA assay with the drop-off ddPCR using 32 identical plasma ctDNA samples gave a concordance of 90.6%. We present a low-cost, highly specific, sensitive and robust assay for blood-based *ESR1* profiling.

**Abstract:**

A considerable number of estrogen receptor-positive breast cancer (ER^+^ BrCa) patients develop resistance to endocrine treatment. One of the most important resistance mechanisms is the presence of *ESR1* mutations. We developed and analytically validated a highly sensitive and specific NaME-PrO-assisted ARMS (NAPA) assay for the detection of four *ESR1* mutations (Y537S, Y537C, Y537N and D538G) in circulating tumour cells (CTCs) and paired plasma circulating tumour DNA (ctDNA) in patients with ER^+^ BrCa. The analytical specificity, analytical sensitivity and reproducibility of the assay were validated using synthetic oligos standards. We further applied the developed *ESR1* NAPA assay in 13 ER^+^ BrCa primary tumour tissues, 13 non-cancerous breast tissues (mammoplasties) and 64 liquid biopsy samples: 32 EpCAM-positive cell fractions and 32 paired plasma ctDNA samples obtained at different time points from 8 ER+ metastatic breast cancer patients, during a 5-year follow-up period. Peripheral blood from 11 healthy donors (HD) was used as a control. The developed assay is highly sensitive (a detection of mutation-allelic-frequency (MAF) of 0.5% for D538G and 0.1% for Y537S, Y537C, Y537N), and highly specific (0/13 mammoplasties and 0/11 HD for all mutations). In the plasma ctDNA, *ESR1* mutations were not identified at the baseline, whereas the D538G mutation was detected in five sequential ctDNA samples during the follow-up period in the same patient. In the EpCAM-isolated cell fractions, only the Y537C mutation was detected in one patient sample at the baseline. A direct comparison of the *ESR1* NAPA assay with the drop-off ddPCR using 32 identical plasma ctDNA samples gave a concordance of 90.6%. We present a low cost, highly specific, sensitive and robust assay for blood-based *ESR1* profiling. The clinical performance of the *ESR1* NAPA assay will be prospectively evaluated in a large number of well-characterized patient cohorts.

## 1. Introduction

Targeted therapies have remarkably changed the treatment of cancer over the last decade [1]. However, almost all tumours acquire resistance to systemic treatment as a result of tumour heterogeneity, clonal evolution, and selection. Estrogen-receptor–positive breast cancer (ER^+^ BrCa) is the most common type of breast cancer diagnosed today. Our understanding of the molecular underpinnings of ER^+^ BrCa have led to new therapies that have substantially improved patient outcomes [2]. However, endocrine-resistant disease remains a leading cause of BrCa mortality [2]. The heterogeneity of metastatic breast cancer (MBC) necessitates novel biomarkers that allow the stratification of patients for treatment selection and drug development, and the early detection of resistance is essential for treatment plans before the onset of metastatic disease [3]. In ER^+^ BrCa, *ESR1* mutations have emerged as a key mechanism of resistance to endocrine therapy [4]. Tissue-based analysis has revealed that *ESR1* mutations have a strong prognostic impact and predictive value of resistance to aromatase inhibitors [4]. Thus, *ESR1* mutation detection is expected to become a prognostic and predictive biomarker in the near future, to be used in clinical practice for hormone-receptor positive (HR+) breast cancer, especially in the metastatic setting [4,5].

‘Liquid biopsy’, based mainly on the analysis of circulating tumour cells (CTCs) and circulating tumour DNA (ctDNA) in the peripheral blood of cancer patients, provides the non-invasive real-time monitoring of tumour evolution and therapeutic efficacy, with the potential to improve cancer diagnosis and treatment, and has received enormous attention because of its obvious clinical implications for personalized medicine [6,7,8]. The molecular characterization of CTCs both at the bulk and at the single cell level [9] has a strong potential to provide information on tumour heterogeneity and to unravel oncogenic alterations related to metastasis or to treatment sensitivity and resistance, and can give valuable information on cancer evolution in real time [9,10,11,12]. The detection of DNA mutations in CTCs in specific genes like *PIK3CA* can define specific subgroups of patients who are suitable for targeted therapy [13,14,15]. In MBC patients with HR+ primary tumours, the lack of ER expression in CTCs could be a possible mechanism of resistance to endocrine therapy [16,17,18]. In parallel, *ESR1* mutation analysis in plasma enables the identification of actionable genomic alterations, monitoring of treatment responses, unravelling therapeutic resistance, and potentially the detection of disease progression before clinical confirmation [19,20].

The implementation of liquid biopsy analysis in clinical practice requires robust, high throughput, and highly sensitive and specific methodologies which are suitable for integration in external quality control schemes. The introduction of very sensitive technologies like droplet digital polymerase chain reaction (PCR) (ddPCR) have enabled the detection of *ESR1* mutations in plasma cell free DNA (cfDNA) and in the CTC-derived DNA of breast cancer patients [21,22,23]. The main limitations of ddPCR is its high cost and the time-consuming nature of the procedure, given that the detection step is separate from the amplification step. Amplification refractory mutation system (ARMS) PCR could also be used as an alternative simple and rapid method for mutation detection [24], but its main limitation is the fact that false positive results may arise because of an inefficient priming on the wild type (wt) sample yielding a late ‘background Cq’ originating from wtDNA [24]. Nuclease-assisted Minor Allele Enrichment using Overlapping probes (NaME-PrO) [25] is an enzymatic approach to the removal of wtDNA from multiple DNA targets selected at will, prior to DNA amplification, after which the current genomic analysis processes remain substantially unchanged. In NaME-PrO, after DNA denaturation, the temperature is reduced to allow the addition of a thermostable double-stranded DNA Duplex-specific nuclease (DSN) and mutation-overlapping oligonucleotide probes that guide the nuclease digestion to the selected wtDNA sequences. We have recently resolved the false positive signals issue derived through classic ARMS-PCR by developing a highly specific and sensitive NAPA (NaME-PrO-assisted ARMS) assay for the simultaneous detection of two *PIK3CA* hotspot mutations (E545K, H1047R) based on a combination of NaME-PrO [25], multiplex ARMS-PCR, and melting analysis [15].

The aim of the present study was to provide a high throughput and reliable analytical tool for blood-based *ESR1* profiling with a low cost, and high sensitivity and specificity. We present the development and analytical validation of a highly sensitive and specific NAPA assay for the detection of four *ESR1* mutations (Y537S, Y537C, Y537N and D538G) and its preliminary application in FFPE tumour tissues, CTCs, and paired plasma ctDNA of ER^+^ BrCa patients.

## 2. Results

### 2.1. Development and Analytical Validation of the ESR1-NAPA (NaME-PrO-assisted) ARMS Assay

The *ESR1-*NAPA assay resolves the false positive signals generated during classic ARMS-PCR, and is based on the combination of: (a) a NaME-PrO step to eliminate the wtDNA, (b) an ARMS-PCR step for detection of the four *ESR1* mutations separately, and 3) a real-time PCR-melting curve analysis step (Figure 1). The experimental conditions were optimized in detail for the annealing temperature, time, and concentration of the primers, buffer, MgCl_2_, dNTPs and bovine serum albumin (BSA) (data not shown). We used synthetic oligonucleotide sequences for each individual *ESR1* mutation as a positive control, and gDNA from HD as a wtDNA control. After the optimization and analytical validation, the developed assay was applied in 13 ER^+^ BrCa primary tumour tissues, 13 non-cancerous breast tissues (mammoplasties) and 64 liquid biopsy samples; these took the form of 32 EpCAM-positive cell fractions and 32 paired plasma ctDNA samples obtained at different time points from 8 ER+ metastatic breast cancer patients, during a 5-year follow-up period. Peripheral blood from 11 healthy donors (HD) was used as a control. We further performed a direct comparison of the *ESR1* NAPA assay with drop-off ddPCR using 32 identical plasma ctDNA samples (Figure 1).

### 2.2. Stability of NaME-PrO (Nuclease-Assisted Minor Allele Enrichment Using Overlapping Probes) Products

The stability of the NaME-PrO products during storage was evaluated by testing the NaME-PrO products from a sample positive for a known *ESR1* mutation that was separated into 10 aliquots. One aliquot was immediately processed (T_0_), whereas five aliquots were stored at −80 °C, and the remaining aliquots were stored at 4 °C. The aliquoted samples were analyzed at five different time points in a total period of 20 days. When the Cq values were compared at day 0 and at each different time point (2-tailed paired *t*-test), our results indicated that the storage of the NAME-pro products at −80 °C for a period of up to 20 days does not significantly affect the results of the mutation analysis (*p* > 0.05); on the contrary, the storage of NAME-pro products at 4 °C, even for one day, affected significantly the results of the mutation analysis (*p* < 0.001) (Appendix A).

### 2.3. ESR1-NAPA Assay Reproducibility

Even though the developed assay was qualitative, we checked its reproducibility based on the derived Tm values using synthetic DNA oligos that were positive for each individual *ESR1* mutation. We used these oligos, diluted in 50 ng WT human genomic DNA at two different concentrations (1% and 5%), in five replicates within the same day. As can be seen in Figure 2, the assay is highly reproducible for all of the mutations tested. Moreover, synthetic oligos for each mutation at a concentration level of 9% were used on 7 different days to check the reproducibility of the assay between the different days. As can be seen in Table 1, the within-day reproducibility of the Tm values for each mutation at mutation allele frequency (MAF) 5% varied from 0.03% to 0.17%, while the between-day reproducibility at MAF 9% varied from 0.12% to 0.46%.

### 2.4. ESR1-NAPA Assay Specificity

The analytical specificity of the *ESR1-*NAPA assay was first evaluated by testing the performance of each individual primer pair and the NaME-PrO probe for all of the *ESR1* mutations. More specifically, one synthetic gene fragment representing each individual mutation (D538G, Y537S, Y537C and Y537N) was used as a target, using all primer pairs and corresponding probes for individual reactions each time. Figure 3 clearly indicates that the *ESR1*-NAPA assay could discriminate each individual mutation specifically, as we did not observe any nonspecific interactions between the primers/probes and the different synthetic oligos. The diagnostic specificity of the *ESR1-*NAPA assay was evaluated by the analysis of 11 plasma cfDNA samples isolated from 11 HD and 13 DNA samples extracted from noncancerous breast tissues (mammoplasties). When the NaME-PrO step was not performed prior to the ARMS-PCR, the false-positive detection of *ESR1* mutations in all of the HD plasma-cfDNA samples was observed; on the other hand, when the NaME-PrO step was implemented before the ARMS-PCR, no false positive results for *ESR1* mutations in any of the healthy donor plasma cfDNA samples were observed (Figure 4).

### 2.5. Analytical Sensitivity

The analytical sensitivity of the *ESR1*-NAPA assay was evaluated using serial dilutions of synthetic oligonucleotide sequences harboring each of the four *ESR1* mutations with wtDNA at various mutation allele frequencies (50%, 9%, 1%, 0.5%, 0.1% and 0%). For the Y537S, Y537N and Y537C hotspot mutations, the developed *ESR1*-NAPA assay could specifically and reliably detect a mutant allelic frequency of 0.1%, while the D538G hotspot mutation was detectable at a mutant allelic frequency of 0.5% (Figure 5). All of the experiments for the evaluation of the analytical sensitivity of the *ESR1*-NAPA assay were independently repeated twice.

### 2.6. ESR1-NAPA Assay: Application in Clinical Samples

#### 2.6.1. Detection of *ESR1* Mutations in Primary Tumours

We applied the developed *ESR1*-NAPA assay to analyze 13 primary tumours from ER-positive breast cancer patients, and from 13 non-cancerous breast tissues (mammoplasties), which were used as the wtDNA control. We detected only the Y537C hotspot mutation in 3/13 (23%) primary ER-positive breast tumour tissues; none of the other three hotspot mutations (D538G, Y537S and Y537N) were detected. We did not detect any *ESR1* mutation in the gDNA samples derived from non-cancerous breast tissues (mammoplasties). The results are shown in Figure 6.

#### 2.6.2. Detection of *ESR1* Mutations in EpCAM-Positive Cell Fractions

The developed *ESR1*-NAPA assay was used to detect *ESR1* mutations in EpCAM-positive cell fractions obtained from eight ER+ MBC patients. EpCAM-positive cell fractions at the baseline were obtained from all patients. For 3 out of these eight patients 23 EpCAM-positive cell fractions were also obtained at different time points during a 5-year follow-up period. Only 1/8 (12.5%) patient sample was found positive; more specifically, the Y537C *ESR1* mutation was detected in patient #7 at the baseline (Figure 7). None of the other three hotspot mutations (D538G, Y537S and Y537N) was detected. All EpCAM-positive cell fractions obtained from HD were negative for *ESR1* mutations.

#### 2.6.3. Detection of *ESR1* Mutations in Plasma ctDNA

The developed *ESR1*-NAPA assay was used to detect *ESR1* mutations in 32 paired plasma ctfDNA samples obtained from the same eight ER^+^ MBC patients. At the baseline, plasma ctDNA samples were available for all eight patients. For three patients, 23 additional samples were obtained at different time points during the follow up period. *ESR1* mutations were not detected in any sample at the baseline (Figure 7). This was expected, as *ESR1* mutations are not detected at the baseline, but only during therapy, as they are derived from resistant clones. It is interesting to note that the D538G mutation was detected in the plasma ctDNA of P#6 consistently at five different time points during the follow-up period. Significantly, this patient later developed clinical resistance to endocrine therapy. In contrast, two patients (P#7, P#8) that were found to be negative for *ESR1* mutations in their plasma ctDNA at all of the time points tested, experienced a clinical objective response to endocrine treatment, and remained free from disease progression at the time of analysis. All of the plasma cfDNA samples obtained from the HD were negative for *ESR1* mutations.

### 2.7. Direct Comparison Between ESR1 NAPA Assay and Drop-Off ddPCR for the Detection of ESR1 Mutations in cfDNA Samples

We further compared the performance of the *ESR1* NAPA assay with droplet digital PCR (ddPCR) using identical samples. More specifically, we analyzed 32 plasma cfDNA samples for the presence of the *ESR1* mutations using the recently-developed drop-off ddPCR [21] and the *ESR1* NAPA assay. As can be seen in Table 2, the developed *ESR1* NAPA assay gave comparable results with the drop-off ddPCR (*p* < 0.001), with a concordance of 90.6% (29/32).

## 3. Discussion

We present here the development and analytical validation of a highly sensitive and specific NAPA assay for the detection of four *ESR1* mutations (Y537S, Y537C, Y537N and D538G) in the primary tumours, CTCs, and plasma-ctDNA of ER^+^ BrCa patients. The assay takes advantage of a pre-PCR step that selectively degrades wtDNA alleles, such that the mutant alleles are preferentially enriched during subsequent PCR using allele-specific primers. The serial combination of two enrichment technologies, NaME-PrO and ARMS, provides high sensitivity and specificity. While alternate mutation enrichment methods such as COLD-PCR (CO-amplification at Lower Denaturation temperature polymerase chain reaction) may also be used [26,27], the ability to degrade wtDNA alleles at the genomic DNA level via NaME-PrO, prior to PCR, overcomes false positives that can be generated via polymerase-introduced errors, thereby boosting specificity. The developed assay gave comparable results with the drop-off ddPCR [21], as the concordance—as evaluated by the parallel analysis of 32 identical plasma ctDNA samples—was 90.6% (29/32).

It is now well established that *ESR1* mutations are selected by prior aromatase inhibitors in advanced breast cancer; the impact of *ESR1* mutations on the sensitivity to standard therapies, assessed in two phase III randomized trials in HR+ advanced breast cancer patients, has clearly shown that *ESR1* mutation analysis in plasma after disease progression under treatment with aromatase inhibitors may help in decision making regarding endocrine-based therapy [28]. The early detection of resistance to endocrine therapy is important, and can be uniquely achieved through the detection of *ESR1* mutations in liquid biopsy samples in real time. This is technically challenging, as CTC are heterogeneous and rare, and the amount of the sample that is available for their analysis is limited [29]. Beyond CTC, the qualitative and quantitative analysis of plasma ctDNA can now be successfully utilized to assess tumour progression and evaluate prognosis, diagnosis and response to treatment in many types of cancer [30,31]. Our group, based on CTC and ctDNA analysis, has recently shown that the epigenetic silencing of *ESR1* through methylation is associated with a lack of response to endocrine treatment [32]. When the utility of the combination of CTCs and ctDNA to predict prognosis in MBC was evaluated, it was concluded that liquid biopsy is an effective prognostic tool [33]. ctDNA analysis in plasma samples from patients with advanced breast cancer has shown that acquired mutations of *ESR1* are a major driver of resistance to fulvestrant/palbociclib combination therapy [34,35,36].

Using the developed highly sensitive and specific *ESR1* NAPA assay, we found that 23% (3/13) of ER+ primary breast tumours harbored the Y537C *ESR1* mutation. This is a relatively high percentage in contrast to studies using ddPCR or NGS (next generation sequencing) that have shown undetectable or extremely low allele frequencies of *ESR1* mutations in primary tumours [37,38,39,40]. Wang et al., using ddPCR, detected *ESR1* mutations—specifically the D538G mutation—in 7.0% (3/43) of ER+ primary tumours at very low mutant allele frequencies [41]. A previous study from the BOLERO trial, using NGS, found *ESR1* mutations in 3% of the primary tumours (6/183) [42]. However, Gelsomino et al. [43] found a prevalence of 12% for Y537N, 5% for Y537S, and 2% for D538G mutation in primary tumours treated with Tamoxifen monotherapy [43]. The higher percentage reported in our study could be explained by the high sensitivity of the *ESR1*-NAPA assay, or could potentially reflect the relatively low number of patients enrolled in the study.

Using the *ESR1* NAPA assay, the D538G mutation was detected in plasma ctDNA during the follow-up period in one patient. This patient was received everolimus and exemestane for a long time, and after the disease progression on this therapy, the D538G *ESR1* mutation was detected in serial samples of their plasma ctDNA. Najim et al. (EROS1 study) identified *ESR1* mutations in patients previously treated with tamoxifen or aromatase inhibitors, revealing a possible correlation between long term aromatase inhibitor therapy and the existence of *ESR1* mutations [44]. On the other hand, we did not detect *ESR1* mutations in plasma ctDNA samples during the follow-up of two other patients with a good response to letrozole and without disease progression.

Our results are consistent with reports which indicate that *ESR1* mutations (especially D538G and Y537S) are associated with more aggressive disease. In EpCAM-isolated cell fractions, only the Y537C mutation was detected in 1 out of 8 (12.5%) patients analyzed at the baseline. Several studies so far have focused on the presence of *ESR1* mutations in CTCs at the bulk or single cell level [39,45,46,47]. The NGS analysis of single CTCs revealed the presence of *ESR1* mutations in 12 patients treated with estrogen deprivation therapy; more specifically, E380Q, Y537C, Y537N, or D538G mutations were found in seven cases, while six novel mutations were also identified. It is remarkable that all of these *ESR1* mutations were not detected in primary tissues, but were only detected in metastases obtained after CTC characterization [39]. In another study, the CellSearch^®^ and DEPArray™ systems were combined in order to isolate single CTCs from ER-positive metastatic breast cancer patients; using MALBAC and Sanger Sequencing, 14 *ESR1* hotspot mutations were detected, and their presence was correlated with endocrine resistance [45]. Paoletti et al. [46] performed a comprehensive mutation and copy number profiling by NGS in 130 genes in CTCs captured from patients with endocrine therapy-resistant MBC. For one patient, who had endocrine therapy-refractory lobular breast carcinoma at the time of research biopsy (whole exome sequencing, WES) and CTC collection (189 days later), they performed a detailed profiling. They detected a heterozygous Y537S mutation in the tissue WES, while Y537S mutations were detected in 26/32 single and pooled CTC samples. Furthermore, in the same patient, *ESR1* A569S mutation was observed in 1 of 32 total CTC samples (single and pooled) [46]. Another group analyzed and compared the mutation profiles in multiple single CTCs and cfDNA isolated from the same blood samples taken from MBC patients using targeted NGS. The ddPCR that was used to validate the selected mutations included *ESR1* Y537S, Y537C, Y537N and D538G. According to the results presented, *ESR1* mutations were present in the single CTCs and cfDNA, but not in the corresponding primary tumour tissues [47]. A perfect concordance of mutations in CTCs and plasma ctDNA is not expected; we have already previously shown—in a direct comparison study between DNAs isolated from Cell-Search cartridges (CTCs) and paired plasma-ctDNA from the same blood draws—a lack of concordance both in early BrCa (48.2%), and in the metastatic setting (66.6%) [14]. According to our results, in most cases, CTCs and plasma ctDNA gave negative results, while in five cases the D538G *ESR1* mutation was detected in cfDNA but not in CTCs, and in one case the Y537C *ESR1* mutation was detected in CTCs but not in cfDNA.

Beyond breast cancer, three potentially-pathogenic *ESR1* mutations were very recently identified in cervical squamous cell carcinoma samples [48]; thus, it would be interesting to use this highly sensitive and specific methodology in cervical cancer samples and potentially pre-malignant cases of endometriosis as well. Very recently, a de Novo *ESR1* Hotspot Mutation was detected in a patient with endometrial cancer treated with an aromatase inhibitor [49], while it has been reported that—in endometrial cancer—*ESR1* mutations are associated with worse outcomes and less obesity [50]. However, the experimental investigation of *ESR1* mutations in endometrial cancer has not been performed. We strongly believe that the present method could be used to study *ESR1* mutations in patients with cervical cancer and endometriosis.

## 4. Materials and Methods

### 4.1. ESR1 Mutation Positive Controls

Four synthetic oligos, each one positive for D538G, Y537S, Y537C and Y537N mutation sites of the *ESR1* gene, were first in-silico designed so that each sequence was represented by a unique gBlock including each specific mutation. These synthetic gene fragments were synthesized as gBlocks by Integrated DNA Technologies (Coralville, IA, USA), and were used as the positive controls for each individual mutation (sequences are available upon request). Lyophilized gBlocks were suspended in Tris/EDTA (Ethylenediaminetetraacetic acid) buffer to a stock solution.

### 4.2. Primers and Probes

For each individual *ESR1* mutation, we designed in silico novel primer pairs using the Primer Premier 5.00 software (Premier Biosoft, CA, USA). An allele-specific ARMS reverse primer was designed for each individual mutation, while a common forward primer was designed for all of the mutations. All of the ARMS primers contain an additional mismatch at one of the last five nucleotides. Α pair of oligonucleotide probes overlapping the target-mutated region (NaME-PrO probes) was designed for each DNA target interrogated for mutations. The NaME-PrO probes contain a polymerase block on their 3΄end in order to prevent polymerase extension in subsequent amplification reactions. All of the primers and probe sequences were designed as previously described [15,25], and are available upon request.

### 4.3. NaME-PrO Step

Prior to the PCR, the NaME-PrO step was applied—as previously described—in order to preferentially digest the wtDNA alleles [15]. At least 10 ng, and up to 50 ng, of gDNA from each sample and overlapping NaME-PrO probes were mixed in 1X DSN buffer (Evrogen, Moscow, Russia) for a total volume of 9.5 μL. The final concentration of the overlapping oligonucleotides’ mix probe was 0.5 μΜ. The samples were placed in a Thermal Cycler (MJ Research PTC-200, MJ Research Inc, Reno, NV, USA) for the denaturation step at 98 °C for 2 min. The samples were then placed on ice for 1 min, and then 0.7 Units of DSN enzyme (Evrogen, Moscow, Russia) was added to the mixture, followed by incubation at 67 °C for 20 min. For DSN inactivation, the samples were incubated at 95 °C for 2 min.

### 4.4. ARMS-PCR and Melting Analysis

The ARMS-PCR and melting analysis were performed in the LightCycler^®^ 480 instrument (Roche, Mannheim, Germany). The amplification reaction mixture for each mutation contained 0.05 U/mL GoTaq Hot Start Polymerase (Promega, Madison, WI, USA), 1X of the supplied PCR buffer, 2.5 mM MgCl_2_, 200 μΜ of each of the four dNTPs, 0.5 μg/μL BSA, 0.3 μΜ of the forward and reverse primers, and 1X LC-Green Plus Fluorescent Dye (Idaho Technology, Salt Lake City, UT, USA) for a final volume of 9 μL. Finally, 1 μL of the NaME-PrO product was added to 9 μL of the reaction mixture. The Real-Time PCR cycling protocol began with an initial denaturation step at 95 °C for 2 min, followed by 45 cycles of: 95 °C for 10 s, 64 °C for 10 s, and 72 °C for 15 s. The real-time fluorescence acquisition was set at 72 °C. The melting analysis cycling protocol included the steps of 55 °C for 1 min, 95 °C for 0 s with a ramp rate of 0.11 °C/s (acquisition mode: continuous), and 40 °C for 1 min. The Real-Time PCR conditions and melting analysis protocols are the same for the detection of all four *ESR1* mutations. Εach experiment was independently repeated twice. In the very few cases that the results were not the same, the experiment was repeated.

### 4.5. Clinical Samples

We applied the developed assay to clinical samples obtained from: (a) 13 ER^+^ BrCa primary tumour tissues and 13 non-cancerous breast tissues (mammoplasties); (b) 9 EpCAM-isolated cell fractions, and the corresponding plasma ctDNA samples obtained from 8 ER+ treatment-naïve MBC patients; (c) 23 EpCAM-isolated cell fractions and the corresponding plasma ctDNA samples obtained from 3 ER+ MBC patients at different time points under endocrine treatment during a 5-year follow-up period; and (d) EpCAM-isolated cell fractions, and the corresponding plasma cfDNA samples obtained from the peripheral blood of 11 healthy donors (HD). It should be mentioned that all of the ER+ MBC patients were receiving hormone therapy throughout the different time points. All of the participating patients signed an informed consent form to participate in the study, and the experimental protocol was approved by the ethics and scientific committees of our institutions. The present study was approved by the Medical Ethical Committee of the General Hospital of Heraklion, Crete, Greece (Ethical Allowance: 8756/23-6-2014).

#### DNA Isolation

In this study, we isolated DNA from three different sources: tissues, CTCs and plasma cfDNA. More specifically:(a)Tissues: all of the primary tumour tissue samples were derived from the primary tumour at the initial diagnosis prior to any systemic treatment (hormonal or chemotherapy). Formalin-fixed paraffin-embedded (FFPE) 10mm tissue sections containing >80% tumour cells were used for the DNA extraction. The genomic DNA was isolated from FFPEs with the High Pure PCR Template Preparation kit (Roche, Mannheim, Germany), according to the manufacturer’s protocol.(b)CTCs: peripheral blood (10 mL) was collected in EDTA tubes. The genomic DNA was extracted from EpCAM^+^ cell fractions using Trizol LS reagent (Invitrogen™, Carlsbad, CA, USA) as previously described [51].(c)Plasma: peripheral blood (10 mL) was collected in EDTA tubes. The plasma was obtained by centrifugation at 530 g for 10 min at room temperature, and a second centrifugation at 2000 g for 10 min, transferred into clean 2 mL tubes, and stored at −70°C. The cfDNA was isolated from the plasma using the QIAamp Circulating Nucleic Acid Kit (Qiagen, Hilden, Germany), as previously described [14]. In all cases, the gDNA concentration was calculated through a standard curve generated from serial dilutions of a wild-type sample with a known DNA concentration (Human Reference DNA Female, Agilent Technologies, Santa Clara, CA, USA). The standards consist of a 10-fold dilution series ranging from 200 ng/μL down to 0.2 ng/μL. RT-qPCR with specific primers for a wtDNA region of the *PIK3CA* gene was performed, as previously described [52].

### 4.6. ESR1 Drop-Off ddPCR: Detection of ESR1 Mutations in Plasma ctDNA Samples

We performed a direct comparison study between our newly-developed *ESR1* NAPA assay and a highly sensitive drop-off ddPCR assay [21]. This assay allowed us to screen the *ESR1* mutations in exon 8, including the Y537S, Y537C, Y537N, D538G and L536R mutations. For this comparison, we used 32 plasma cfDNA samples and analyzed them by both methodologies. These 32 samples were analyzed in our QX200 Droplet Digital PCR System (Bio-Rad Laboratories, Hercules, CA, USA) following the drop-off ddPCR protocol, using the published primers and TaqMan^®^ hydrolysis probes (ThermoScientific LSG, Walttam, MA, USA) exactly as previously described [21]. In particular, we used a probe complementary to the WT sequence of the altered region (Hotspot probe, VIC-labeled), which detects all of the mutations contained in codons 536, 537, and 538, and a reference probe designed over a nonmutated region within the same amplicon (REFex8 probe, FAM labeled). Droplets containing WT alleles were double positive (FAM^+^/VIC^+^), while the mismatch due to the mutations leads to a VIC signal decrease (FAM^+^/VIC^low^). Briefly, a 20 μL reaction mixture containing 10 μL of 2× ddPCR Supermix for probes (No dUTP) (Bio-Rad Laboratories, Hercules, CA, USA), 250 nM of each primer, 450 nM of each probe, and 8μL of the cfDNA samples in a reaction volume of 20 μL, adjusted with sterile water, was used according to the published assay [21]. After the droplet generation, the entire volume of the PCR mix was transferred for thermal cycling under the following program: 95 °C 10 min, 94 °C 30 s and 58 °C 60 s (40 cycles), 98 °C 10 min, and an infinite hold at 12 °C. For each run, negative (no DNA) and positive controls (synthetic DNA oligos) were included. The data were exported to comma-separated value files, and were analyzed with the program developed by Attali and colleagues specifically for this type of ddPCR assay [53].

### 4.7. Statistical Analysis

The statistical analysis was performed using IBM SPSS Statistics, version 23.0 (SPSS Inc., Chicago, IL, USA). A *p* value < 0.05 was considered statistically significant. The graphics were generated with MS Excel 2010. A 2-tailed paired *t*-test was used to determine the stability of the NaME-PrO products.

## 5. Conclusions

In this study, we presented a reliable, highly specific and sensitive robust assay for the detection of *ESR1* mutations in liquid biopsy samples. The developed assay is fast, with a comparable sensitivity to ddPCR, but with a much lower cost, and can thus be used as a fast and relatively-cheap screening method to classify patients as positive or negative for *ESR1* mutations in liquid biopsy samples. A limitation of this assay is that it is not quantitative, and thus cannot be used to evaluate the mutational load in plasma. The clinical performance of the *ESR1* NAPA assay for blood-based *ESR1* profiling will be prospectively evaluated in a large number of well-characterized patient cohorts.

## Figures and Tables

**Figure 1 cancers-13-00556-f001:**
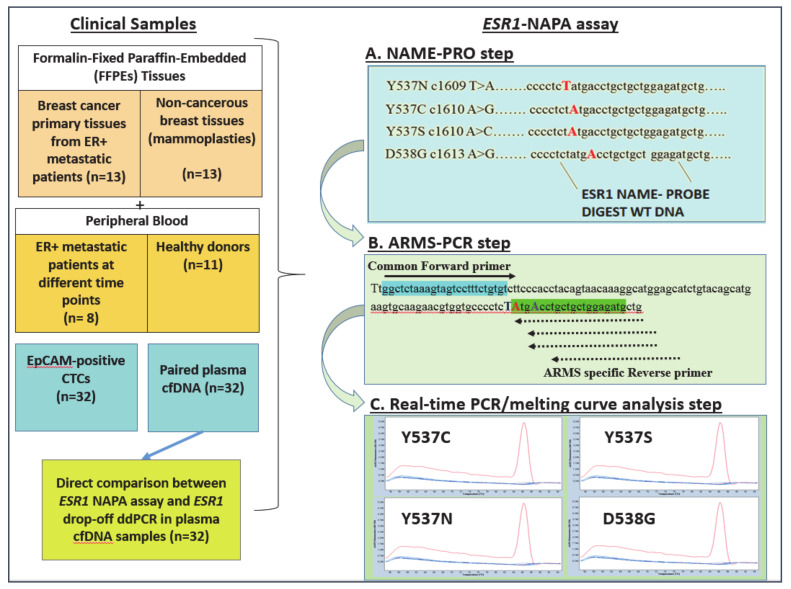
Outline of the experimental procedure.

**Figure 2 cancers-13-00556-f002:**
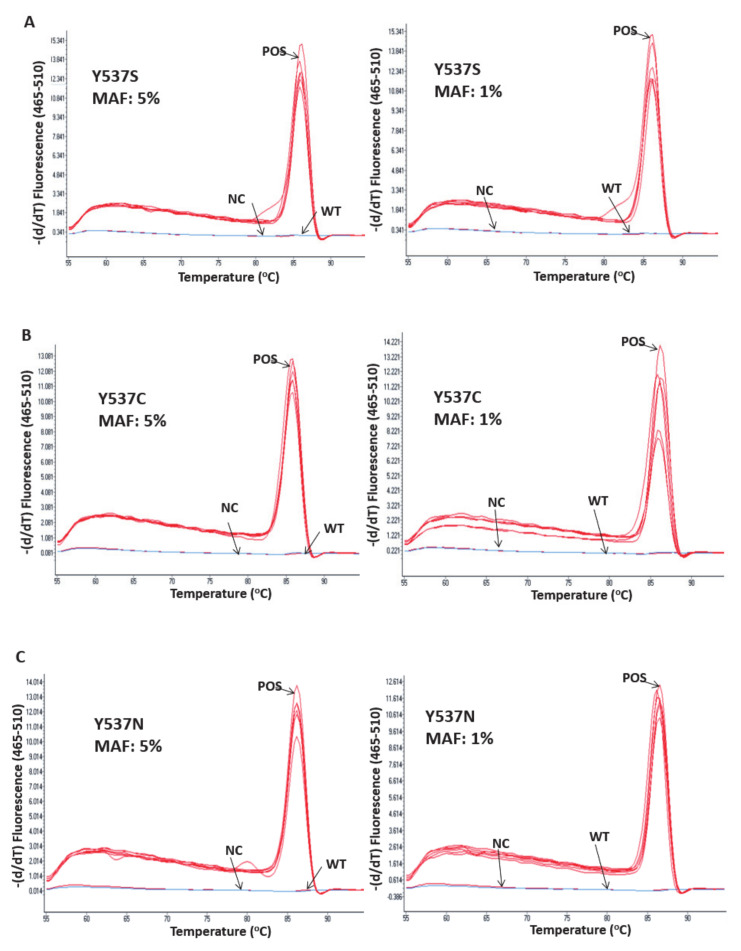
Within-day reproducibility of the *ESR1*-NAPA assay using synthetic DNA oligos for D538G, Y537S, Y537C and Y537N mutations at 1% MAF and 5% MAF (five replicates), (**A**) Y537S, (**B**) Y537C, (**C**) Y537N, (**D**) D538G. POS: the positive control was a synthetic oligonucleotide sequence harboring each of the four *ESR1* mutations (9% MAF); NC: PCR negative control; WT: wild type.

**Figure 3 cancers-13-00556-f003:**
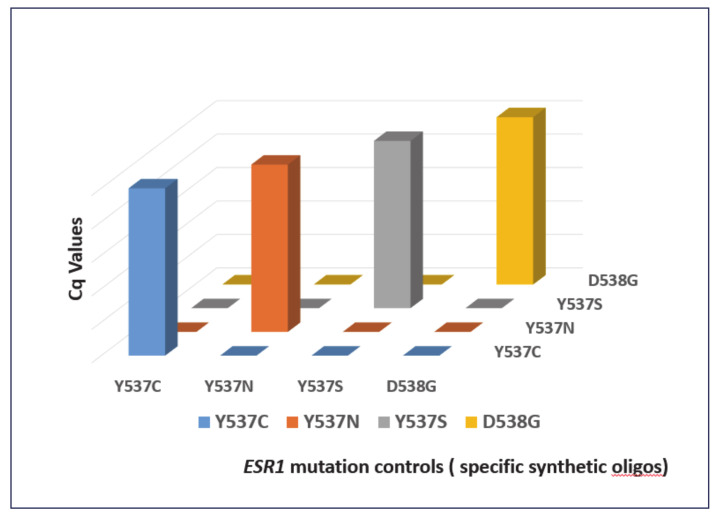
Analytical Specificity of the *ESR1* NAPA assay: blue: Y537C; red: Y537N; grey: Y537S; yellow: D538G. All of the experiments were performed in duplicate.

**Figure 4 cancers-13-00556-f004:**
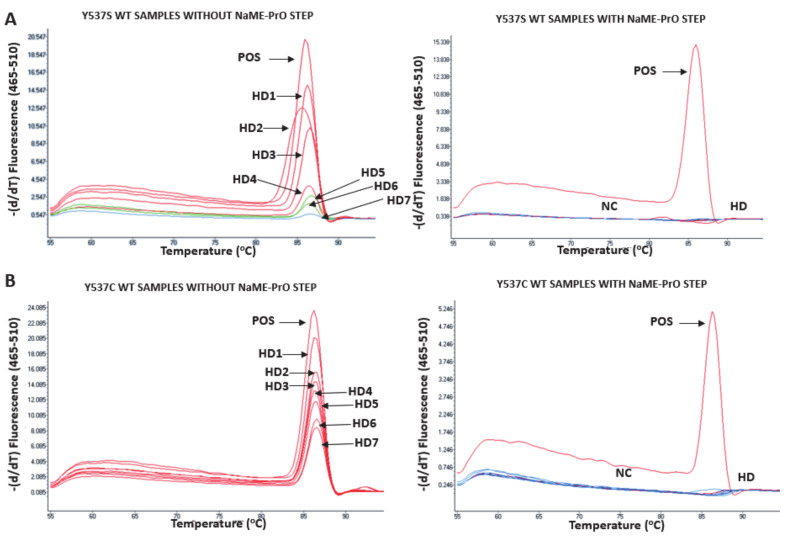
Diagnostic specificity of the *ESR1* NAPA assay. Analysis of plasma-cfDNA samples from 7 healthy donors (HD) with and without NAME-PrO assay treatment prior to the ARMS-PCR. As a positive control (POS), we used synthetic oligonucleotide sequences harboring each of the four *ESR1* mutations with WT gDNA (genomic DNA) at 9% mutation allele frequency. With the NAME-PrO assay treatment prior to the ARMS-PCR, none of the *ESR1* mutations were detected in the HD plasma-cfDNA samples (red). NC: PCR negative control (blue). (**A**) Y537S mutation, WT samples without NAME-PrO step, and with NAME-PrO step, (**B**) Y537C mutation, WT samples without NAME-PrO step, and with NAME-PrO step, (**C**) Y537N mutation, WT samples without NAME-PrO step, and with NAME-PrO step, (**D**) D538G mutation, WT samples without NAME-PrO step, and with NAME-PrO step.

**Figure 5 cancers-13-00556-f005:**
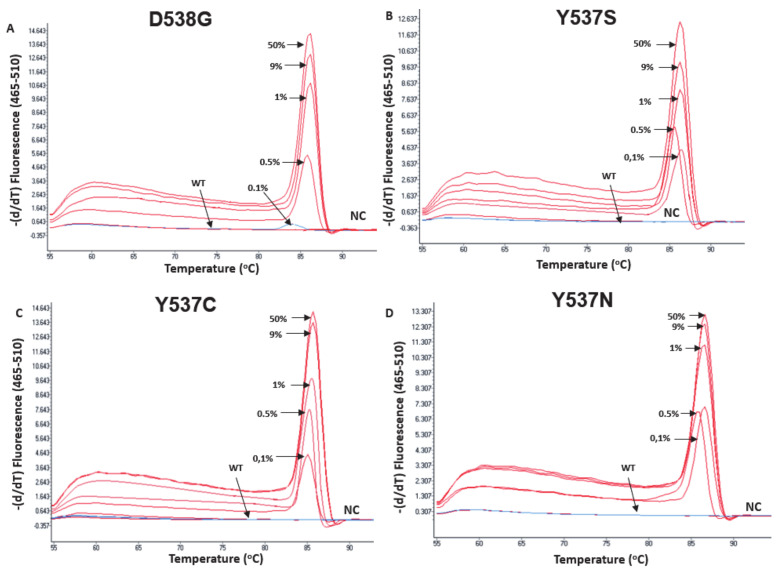
Analytical sensitivity of the *ESR1* NAPA assay in samples prepared by mixing known concentrations of mutated synthetic oligos with wtDNA. The mutation allele frequencies (MAFs) tested were 50%, 9%, 1%, 0.5%, 0.1% and 0%. WT: wild type; NC: PCR negative control. (**A**): D538G, (**B**) Y537S, (**C**) Y537C, (**D**) Y537N

**Figure 6 cancers-13-00556-f006:**
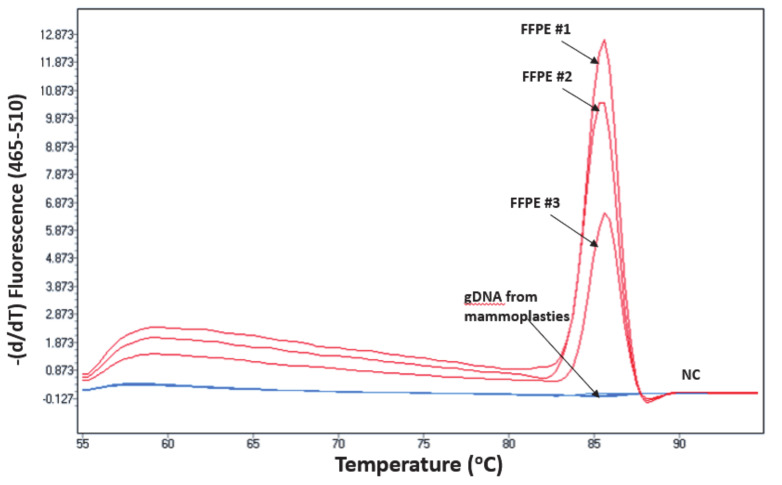
*ESR1* NAPA assay: detection of the Y537C mutation in primary ER-positive breast tumour tissues (FFPEs) and mammoplasties. NC: PCR negative control.

**Figure 7 cancers-13-00556-f007:**
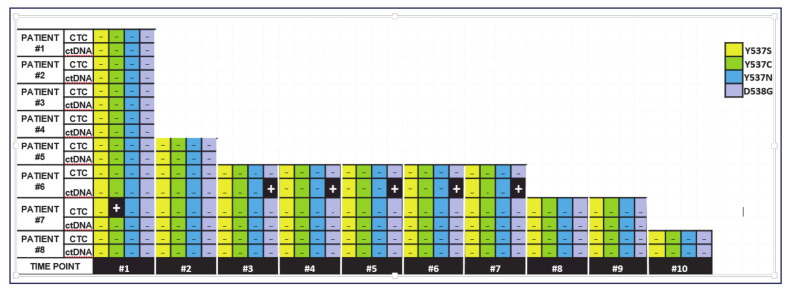
*ESR1* NAPA assay application in liquid biopsy samples.

**Table 1 cancers-13-00556-t001:** Reproducibility of the *ESR1* NAPA assay using synthetic oligos for each *ESR1* mutation.

*ESR1* Mutation	Within-Day Reproducibility (n = 5) (MAF *, 5%)		Day to Day Reproducibility (n = 7) (MAF: 9%)	
	**Tm* (°C)**	**CV*%**	**Tm (°C)**	**CV%**
Y537S	85.97 ± 0.14	0.17	85.51 ± 0.40	0.47%
Y537C	85.83 ± 0.07	0.08	85.59 ± 0.14	0.16%
Y537N	86.23 ± 0.03	0.03	85.82 ± 0.10	0.12%
D538G	85.90 ± 0.15	0.17	85.58 ± 0.19	0.23%

* MAF: Mutation Allele Frequency, Tm: Melting Temperature, CV: Coefficient of Variation.

**Table 2 cancers-13-00556-t002:** Comparison between the developed *ESR1* NAPA assay and the drop-off ddPCR for 32 identical cfDNA samples.

*ESR1* Drop-Off ddPCR				
		**+**	**−**	**Total**
***ESR1*** **NAPA assay**	**+**	5	0	5
	**−**	3	24	27
	**Total**	8	24	32
**Concordance:**	**(29/32) 90.6%, (Chi-Square Test, *p* < 0.001)**			

## Data Availability

The data presented in this study are available on request from the corresponding author.

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
