# Peer review of "ESR1 NAPA Assay: Development and Analytical Validation of a Highly Sensitive and Specific Blood-Based Assay for the Detection of ESR1 Mutations in Liquid Biopsies"

_cancers, 2021, doi:10.3390/cancers13030556_

Round 1

Reviewer 1 Report

The manuscript entitled “ESR1 NAPA Assay: A Highly Sensitive and Specific Blood-Based Assay for the Detection of ESR1 Mutations in Liquid Biopsies” present the development and analytical validation of a qPCR-based assay for the detection of four ESR1 mutations (Y537S, Y537C, Y537N and D538G). Next the authors show its application on breast cancer patient samples that are either gDNA extracted from fixed tumor tissues (FFPE), gDNA extracted from purified circulating tumor cells (CTC) or cell free DNA (cfDNA) isolated from blood. 

This manuscript is a new submission after revisions. I reviewed this article in its first version, and my opinion at that time was that the experiments performed were not robust enough to claim for an analytical validation of the molecular test.

The authors have well improved the description of the study, and the samples they used. These points are now clear. The authors also added a comparison of the four mutations detection by the qPCR based assay and by another reference technique, the ddPCR, for 32 cfDNA samples.

To be able to claim for a robust validation of this molecular assay, the authors should perform the following steps on the final target material that are the genomic DNA extracted from the 3 sources described above (FFPE, cfDNA and CTC).

  1. Test the reproducibility (need several experiments with several operators)
  2. Test the repeatability
  3. Test the sensitivity (need another measurement of the same samples by another reference technique to detected true positif, true negative, false positive and false negative)
  4. Test the specificity (need another measurement of the same samples by another reference technique to detected true positif, true negative, false positive and false negative)
  5. Determine the level of blank and the level of detection (LOB and LOD).

These tests were partially performed on synthetic oligos containing the four ESR1 mutations. The test 1 and 5 were missing on these oligo.

But these test were not performed on the target material, at the exception of the test 3 and 4 on cfDNA that were compared to ddPCR, on 32 cfDNA.

Synthetic oligo have a much better quality than gDNA extracted from FFPE tissues, or cfDNA. As a consequence, the PCR efficacies and threshold for positive signals of the ARMS PCR will probably be very different on those degraded gDNA. As a consequence, the validation of the assay need to be performed on the target clinical material.

 The main claim of this article remains thus unreached. The claim should be revised to describe a “proof of concept” of the technique, at most.

Moreover, the results of the 5 steps for a validation assay should be described as quantitative values and not as qualitative figures as currently shown in the manuscript. Sensibilities and specificities of the assay on the 32 cfDNA should be written as a result.

Minor changes:

Line 106 : FFPES instead of FFPE tumor tissues

Line 137 : synthetic DNA oligos positive for each individual ESR1 mutation, at two different concentrations (1% and 5%). Please indicate in what are re-suspended the 1% and 5% mutated synthetic oligos? In human genomic DNA? from human? Extracted from FFPE tissue or CTC or cfDNA?

Reviewer 2 Report

Dr. Stergiopoulou and her group report the development and validation of a novel assay for the detection of ESR1 mutants in HR+ metastatic breast cancer. ESR1 mutations were described in tissues samples from patients with HR+ MBC exposed to aromatase inhibitors. After initial observations, several retrospective studies evaluating both, tissue and plasma demonstrated the important prognostic and predictive value of acquired ESR1 mutations. Liquid biopsy, particularly ctDNA has shown to represent ideal tool for monitoring molecular evolution of advanced solid tumors, including breast cancer. The use of ddPCR and NGS for plasma analysis confirmed the ability to detect and monitor ESR mutations during therapy. Unfortunately, the cost associated with such testing reduce large-scale application in patients care, therefore there is need for development of targeted, low-cost diagnostic testing for detecting most common variants of ESR1 during tumor evolution.

Prof. Lianidou’s laboratory outline the development of the new NaME-PrO-assisted ARMS (NAPA) assay to detect hot-spot ESR1 mutations (Y537S, Y537C, Y537N and D538G). The testing and validation include tissue samples from primary tumor, normal individuals and plasma and CTCs from 8 HR+ metastatic breast cancer patients. Samples were analyzed for stability, reproducibility and, most importantly specificity and sensitivity. The sequence of NaME-PrO-assisted ARMS (NAPA) eliminated false positive results and the lower limit of detection of those variants was between 0.1% and 0.5%.

ESr1 mutations are rare in primary disease and, accordingly were detected only in one case. Detection of ESR1 mutation was then evaluated in CTCs (EpCAM-isolated) and also, demonstrated rare event (one case at baseline).  When analyzing serial plasma samples, the investigators found evidence of ESR1 mutations in the follow-up period of one case as indication of evolution. The final step consisted in the validation with ddPCR demonstrating a concordance of >90%. The technology is robust and shoud be applied to larger cohorts to address clinical validity in clinical setting because we have much to learn from this limited experience. There are some important considerations to be addressed:

  1. The authors used primary tissue from 13 patients with metastatic breast cancer but, no metastatic tissue samples was evaluated to demonstrate the presence of ESR1 The assessment of metastatic disease at time of recurrence with standard biomarkers and molecular testing (e.g. ESR1 genomic analysis) represents the goal standard and would have provided a more solid assessment of concordance.
  2. I would recommend a table with summary of patients’ characteristics and endocrine treatments to assess predictive value of ESR1, particularly relevant for patient #6.
  3. The timing of baseline and follow-up CTCs and plasma specimens would be important to clarify in order to understand the relation between clinical status (primary vs. metastatic) and ESR1 mutational status.

Reviewer 3 Report

Review paper Cancers 1042918

Title: ESR1 NAPA Assay: A Highly Sensitive and Specific Blood-Based Assay for the Detection of ESR1Mutations in Liquid Biopsies

The manuscript titled " ESR1 NAPA Assay: A Highly Sensitive and Specific Blood-Based Assay for the Detection of ESR1 Mutations in Liquid Biopsies" ,byStergiopoulou D. et.al, developed an NaME-PrO-assisted ARMS (NAPA) assay. The assay is proposed for the detection of four ESR1 mutations (Y537S, Y537C,Y537N and D538G) in circulating tumour cells (CTCs) and paired plasma circulating tumour DNA (ctDNA) in patients with ER+ BrCa.

Here are some major issues:

  1. in order to remove artefacts, especially those including deamination that results in C:G>T:A substitutions, as reported by Hofreiter M. et al (DNA sequences from multiple amplifications reveal artefacts induced by cytosine deamination in ancient DNA. Nucleic Acids Res 2001; 29(23): 4793-9) a predictive regression model with true/false positive mutation could be built, based on a subset of variants classified with high confidence as true and false positives.
  2. to evaluate the specificity of an analytical assay to be applied to humans, it is necessary to identify physiological, paraphysiological and / or pathological conditions characterized by a biological scenario similar or nearly to the disease in which the biomarker is analyzed. For ESR1 mutations, endometriotic disease is the pathology of choice to evaluate the reliability of the test, in term of specificity, for the following reasons:
  3. Endometriosis is often considered a subclinical disease and therefore underestimated even in a possible sampling of "apparently" healthy subjects. Moreover, it can be exasperated by the treatment with tamoxifen, as reported in the article by Peter G. Rose et al. (VOLUME 183, ISSUE 2, P507-508, AUGUST 01, 2000 Exacerbation of endometriosis as a result of premenopausal tamoxifen exposure. DOI: https: //doi.org/10.1067/mob.2000.105966).
  4. It is a pathology associated with the risk of breast cancer and it coexists with that of breast cancer associated with the risk of first-degree relatives of breast cancer (OR = 5.69 (95% CI, 2.4-13.3), P <0.001) in many patients.
  5. Finally, its pathogenesis is strongly linked to a polymorphism, precisely the one for estrogen receptors (Li Y, Liu F, Tan SQ, Wang Y, Li SW. Estrogen receptor-alpha gene PvuII (T / C) and XbaI (A / G) polymorphisms and endometriosis risk: a meta-analysis. Gene. 2012 Oct 15; 508 (1): 41-8. doi: 10.1016 / j.gene.2012.07.049. Epub 2012 Aug 4. PMID: 22890138).

Therefore, I suggest to associate the healthy cohort (11HD) to an equal cohort of women diagnosed with endometriosis on which the analytical assay should be repeated on ctDNA to complete the validation phase related to the specificity of the test to reported in sections 2.4 and 2.6.3 of the maniscript.

  1. Figure 7 shows the negativity of some CTC samples with respect to the ctDNA and viceversa, which the authors report clearly and transparently. The question is: why in the same patient a different biological matrix, which however has the same origin, gives different results? This condition shows a problem in the analytical method standardization or in the collection method of one of the two matrixes. This analytical assay is used on both CTCs and circulating DNA. However, of all the CTCs present in the bloodstream, the authors select only the subset of CTCs-EPCAM positive. It is therefore plausible that the negative result recorded on the CTCs samples may actually be a false negative. On the other hand, could the negative results for ctDNA and positive for CTCs samples be interpreted as false positives? Cell is self-referential, when identified with specific breast markers, for this assay, should overcome this problem, because the selection from the blood of a mammary epithelium cell removes any doubt about the specificity of the result obtained. Otherwise, with positive results on ctDNA and negative on CTC, is it definable as a real positive case? The analysis of correspondent tumour biopsy, as reported by the authors themselves, is not a good reference for the presence of this mutation linked to the metastatic behaviour of breast disease. The mutated DNA molecule does not carry with it any identifying sign of its tissutal origin, if not the mutation itself. Could ESR1 mutation also be present in circulating epithelial cell derived from another tissue source, i.e. ovarian, in a woman with exasperated tamoxifen endometriosis?

I believe that the analysis presented by these authors is very interesting and more reliable if it can be done using a combination of ctDNA and CTCs.

A summary receiver operating characteristic curve (ROC curve) and area under the curve (AUC) will be used to summarize the overall test on specificity performance by using single and combined circulating biomarkers will be appreciated.

Minor issues

Please, fix several minor typos presented in the text

In conclusion: In the current format, the article should not be published.

Round 2

Reviewer 1 Report

The authors did not modify significantly their manuscript. The main claim of this article remains thus unreached: the analytical validation of the assay is not robust enough, for the reasons exposed in my last reviewing.

Thus, I still suggest to present this study as a “proof of concept”, rather than as an “analytical validation” of this assay.

Author Response

please see attached letter

Author Response

please see attached letter
